# Nanohydroxyapatite as a Biomaterial for Peripheral Nerve Regeneration after Mechanical Damage—In Vitro Study

**DOI:** 10.3390/ijms22094454

**Published:** 2021-04-24

**Authors:** Benita Wiatrak, Paulina Sobierajska, Marta Szandruk-Bender, Paulina Jawien, Maciej Janeczek, Maciej Dobrzynski, Patrycja Pistor, Adam Szelag, Rafal J. Wiglusz

**Affiliations:** 1Department of Pharmacology, Wroclaw Medical University, Mikulicza-Radeckiego 2, 50-345 Wrocław, Poland; benita.wiatrak@umed.wroc.pl (B.W.); marta.szandruk@umed.wroc.pl (M.S.-B.); paulina.jawien@umed.wroc.pl (P.J.); adam.szelag@umed.wroc.pl (A.S.); 2Institute of Low Temperature and Structure Research, Polish Academy of Sciences, Okolna 2, 50-422 Wroclaw, Poland; 3Department of Biostructure and Animal Physiology, Wrocław University of Environmental and Life Sciences, Norwida 25/27, 50-375 Wrocław, Poland; maciej.janeczek@upwr.edu.pl (M.J.); 110489@student.upwr.edu.pl (P.P.); 4Department of Pediatric Dentistry and Preclinical Dentistry, Wroclaw Medical University, Krakowska 26, 50-425 Wroclaw, Poland; maciej.dobrzynski@umed.wroc.pl

**Keywords:** nanomaterials, nanohydroxyapatite, lithium ions, europium ions, nerves regeneration

## Abstract

Hydroxyapatite has been used in medicine for many years as a biomaterial or a cover for other biomaterials in orthopedics and dentistry. This study characterized the physicochemical properties (structure, particle size and morphology, surface properties) of Li^+^- and Li^+^/Eu^3+^-doped nanohydroxyapatite obtained using the wet chemistry method. The potential regenerative properties against neurite damage in cultures of neuron-like cells (SH-SY5Y and PC12 after differentiation) were also studied. The effect of nanohydroxyapatite (nHAp) on the induction of repair processes in cell cultures was assessed in tests of metabolic activity, the level of free oxygen radicals and nitric oxide, and the average length of neurites. The study showed that nanohydroxyapatite influences the increase in mitochondrial activity, which is correlated with the increase in the length of neurites. It has been shown that the doping of nanohydroxyapatite with Eu^3+^ ions enhances the antioxidant properties of the tested nanohydroxyapatite. These basic studies indicate its potential application in the treatment of neurite damage. These studies should be continued in primary neuronal cultures and then with in vivo models.

## 1. Introduction

The demand for biomaterials is constantly growing due to the increasing life expectancy, which leads to an increase in the number of patients with skeletal system defects. Large defects often require the simultaneous regeneration of bone and nervous tissue. Therefore, the current challenge is to obtain such implantable materials that enable innervation of the healing bone. Materials implantable into the human body must be biocompatible and preferably bioactive, and over time, after tissue reconstitution, also resorbable [1]. One of the implantation materials is based on calcium phosphates, the so-called hydroxyapatite (HAp), which may be of synthetic origin or derived from natural sources. These materials are characterized by chemical and mineralogical similarities to inorganic substances of bones and teeth. When analyzing the chemical composition of the skeletal system, it was found that 30% of bones are made of organic matter and 70% of inorganic matter or water, including hydroxyapatite (which accounts for about 65% of bone mass) [2,3,4]. The isolated crystals of HAp from natural bones are deficient in calcium, possess rod-like or needle-like morphology in nanoscale, and are poorly crystalline [5]. From a medical point of view, poorly crystalline or even amorphous hydroxyapatite is unfavorable for bone grafting due to its very fast dissolution rate. Many studies found out that an ideal material for orthopedics was HAp with a Ca/P molar ratio close to that of stoichiometric value equal to 1.67 [3,6,7,8]. It was reported that the required resorbability of hydroxyapatite can be achieved by controlling its degree of crystallinity to 60–70% and reducing its particle size to the nanolevel [9]. Synthetic nanoapatite (nHAp) shows a high degree of biocompatibility, osteoinductivity, and osteointergrality [10,11,12]. Due to the similarity to bone, no cytotoxic effect is observed after the implantation of synthetic hydroxyapatite. Hydroxyapatite is used in several forms, i.e., nanoparticles, nanocoatings, porous scaffolds or even composite scaffolds (as a combination of ceramic and polymer) [12]. The use of hydroxyapatite as a coating improves the biological properties of the medical device. The addition of HAp to natural polymers allows for better mechanical and biological properties of the designed product [12].

Numerous ionic substitutions are present in the crystal lattice of biological apatites, such as Mg^2+^, Na^+^, K^+^, Sr^2+^, Zn^2+^, Cl^−^, F^−^, and CO_3_^2^^−^. These ions are important in regulating biochemical reactions related to bone metabolism—they affect the activity of enzymes. Therefore, substitution of synthetic HAp with various ions is an innovative approach in the design of materials for biological applications. Lithium (Li^+^) seems to be a good candidate as a therapeutic agent for tissue engineering purposes. It was found that the addition of lithium ions (Li^+^) at the sintering process improves the density, microhardness, and microstructure of ceramics [13]. The degradation or bioresorption property of Li^+^-doped hydroxyapatite was investigated by Wang et al. [14] in the context of the possible use of studied biomaterial for bone regeneration. Doping with Li^+^ ions had a better effect on the growth of osteoblast cells. Our previous studies [15] confirmed the literature results regarding the pro-proliferative and anti-apoptotic effects of lithium on progenitor cells. We showed that lithium ions were effectively released from the apatite matrix, which had a beneficial effect on stem cells, and thus we found a novel Li^+^ ion delivery method based on nHAp application.

Damage to neurons and axons can cause sensory and motor dysfunctions. During bone injuries, damage to blood vessels and peripheral nerves is observed [16]. Axon scarring inhibits the regenerative growth of the nerve. Effective treatment of mechanical injuries of the spinal cord and the peripheral nerves remains one of the major clinical challenges in modern medicine. It is well known that the regenerative capacity of nerve cells is limited, and that mainly nerve regeneration is observed in olfactory glial cells [17]. However, in the case of minor injuries of peripheral nerves, various biomaterials are used to treat nerve injuries [18,19]. The promotion of adult neurogenesis by various chemical agents may offer a potential tool for repairing neurons damaged due to traumatic injuries. Ideal material for nerve regeneration should have physicochemical properties similar to the nerves [20,21]. The neuroregenerative activity of nHAp has been demonstrated in various scaffolding materials, e.g., in type I collagen gel, nanocellulose hydrogel, and carbon nanotubes [16,21,22,23,24]. Nanohydroxyapatite can also be used as a coating for other materials that are themselves neurotoxic, e.g., carbon nanotubes, which improve nerve regeneration by directing axon growth [23]. Coverage with nHAp has been shown to significantly increase axon migration and elongation [16,23]. Moreover, nHAp did not neutralize the electrophysiological activity of neurons, as Liu et al. demonstrated in primary cultures of rat neurons [23]. Currently, nanotubes made in 3D printing technology from combined materials containing hydroxyapatite are also being studied [24,25]. In vivo studies have shown that nHAp has a beneficial effect on nerve regeneration in rats after prior spinal distension. Stretching causes swelling and hemorrhage in the spine, and nHAp treatment has reversed bleeding and edema [16]. Studies of the neuroregenerative activity of nHAp dispersed in the type I collagen hydrogel were carried out on Wistar rats after crushing the sciatic nerve, which confirmed its stronger neuroregenerative properties than a collagen-only hydrogel [22]. The neuroinductive properties of nHAp were also observed in nanocellulose hydrogel in in vitro models.

In this study, we proposed lithium ions, incorporated into the nanohydroxyapatite structure, as an active agent that can affect the regeneration of nervous tissue. The cellular mechanism by which Li^+^ influences the regeneration of nervous tissue became the subject of many reports. For example, Su et al. [26] showed that lithium stimulated the survival, proliferation, and differentiation of neural progenitor cells used to treat the damaged spinal cord. Vazey and Connor [27] demonstrated that lithium chloride (LiCl) may improve cell transplantation efficiency in rats and accelerate sensorimotor function recovery. Makoukji et al. [28] showed that LiCl applied in mice after facial nerve crush injury caused the significant stimulation of myelin gene expression, restoration of myelin structure, and thereby the recovery of whisker movements. Additionally, Zhu et al. [29] demonstrated that lithium ions can increase the efficacy of neural progenitor cells by increasing neurogenesis and reducing astrogliogenesis.

This work focuses on the effect of nHAp on the regeneration of mechanical damage in peripheral nerve injuries. The study was carried out on the PC12 and SH-SY5Y cell lines, which, after differentiation with nerve growth factor or retinoic acid, respectively, are widely used as a research model in neurobiology [30]. These differentiated cells are characterized by microscopically visible neurites and express dopamine, noradrenaline, and catecholamine. We decided to check whether the Li^+^- and Li^+^/Eu^3+^-doped nanocrystalline hydroxyapatite obtained by our group affects nerve regeneration. One of the more interesting aspects we studied was investigating the effect of europium ions (Eu^3+^) on peripheral nerve cells. The toxicity of the Eu^3+^ ion and the other lanthanide ions remains a matter of dispute. Our previous research has shown that Eu^3+^ ions promote the proliferation of adipose-derived stem cells [15,31,32]. Moreover, the luminescent properties of Eu^3+^ ions shed promising light on their applications in bio-imaging. Lanthanide ions are characterized by a relatively long and photostable emission and could be used in the bio-imaging of implants in real time [33]. Moreover, our previous results have revealed that the lithium ions were located in the nHAp: Li^+^/Eu^3+^ system as charge compensators, showing the enhancement of the luminescent efficiency of Eu^3+^ ions [34].

## 2. Results and Discussion

### 2.1. Physicochemical Characterization of the Studied Materials

#### 2.1.1. Structure Analysis

The formation of the hydroxyapatite doped with Li^+^ and co-doped with Li^+^/Eu^3+^ ions was confirmed by the X-ray diffraction (XRD) technique. As can be seen in Figure 1a, in the case of both materials, the diffraction peaks are fully overlapped with those of Ca_10_(PO_4_)_6_(OH)_2_ originated from the standard diffraction file (ICSD–26204), and no additional phase can be identified. The XRD patterns show the main diffraction peaks of the apatite phase with hexagonal structure (P6_3_/m, No. 176) at 2θ (hkl) equal to 25.85° (002), 31.74° (121), 32.15° (112), 32.87° (300), 34.03° (202), 39.77° (310) and other minor peaks in the 2θ range of 40–60°. Depending on the type of dopant, the average crystallite size was 47 nm for the hydroxyapatite doped with Li^+^ and 39 nm for the nHAp co-doped with Li^+^ and Eu^3+^. Table 1 shows the calculated lattice parameters (a = b and c) and cell volume (V) for both materials, indicating cell growth induced by Li^+^ ions and, on the other hand, cell contraction after further Eu^3+^ substitution. The calculated cell parameters were in good correlation with the results observed for the (0 0 2) plane shift. In the case of nHAp: Li^+^/Eu^3+^, the peak position shift towards higher 2θ angles is associated with a decrease in lattice parameters. The observed dependencies are the consequence of the formation of charge-compensating defects caused by aliovalent cation doping. Based on the ionic radii values given by Shannon [35], replacing Ca^2+^ (1.18 Å for CN_9_ and 1.06 Å for CN_7_; CN: Coordination Number; Figure 1b) with smaller Eu^3+^ (1.12 Å for CN_9_ and 1.01 Å for CN_7_) and much smaller Li^+^ (0.92 Å for CN_9_) should lead to a reduction in the unit cell. In fact, the opposite result is observed for the sample doped only with Li^+^ ions. Incorporation of the additional Eu^3+^ ions into the hydroxyapatite structure caused a marked reduction in the nHAp cell. It is well documented in our previous research [34,36] that the substitution of the divalent calcium ions by trivalent rare earth ions (RE^3+^) leads to charge imbalance, and some defects in the HAp lattice are created to retain electro-neutrality. Our studies have shown that Li^+^ acts as an effective charge compensator, which affects site occupation (Figure 1b) and contributes to the enhancement of RE^3+^ luminescence from the apatite matrix.

Variations of crystallinity degree and microstrain with Li^+^ and Eu^3+^ doping are shown in Table 1. The observed decrease in strain (in-plane and out-of-plane) for nHAp: Li^+^ is attributed to the increase in crystallinity (χ_c_). For this sample, the reduction in full width at half maximum (FWHM) of the XRD diffraction peaks indicates that lithium acts as a hydroxyapatite crystallization enhancer [37]. The diffraction peaks from hydroxyapatite doped only with the Li^+^ ions are slightly sharper than for co-doped nHAp. Moreover, it is seen that (112) and (300) reflections for nHAp: Li^+^ are more split than for nHAp: Li^+^/Eu^3+^.

In order to obtain more detailed information about the structure of obtained materials, FT-IR spectroscopy was applied. All the vibration mode positions confirmed hydroxyapatite formation (Figure 2) [39]. The IR spectra indicated the typical hydroxyapatite absorption bands at about 631 cm^−1^ and 3573 cm^−1^ belonging to the vibrational (*ν_L_*) and stretching (*ν_S_*) modes of the ‒OH group, respectively. The typical absorption bands of the phosphate PO_4_^3−^ groups were located at about 472 cm^−1^ (the doubly degenerate *δ_2_* bending); 560 cm^−1^ and 601 cm^−1^ (the triply degenerate *δ*_4_ bending); 961 cm^−1^ (the symmetric non-degenerate stretching *v*_1_ vibrations); 1020 cm^−1^ and 1089 cm^−1^ (the asymmetric triply degenerate stretching *v*_3_ vibrations) [15].

#### 2.1.2. Particle Size, Morphology and Surface Studies

The nano-nature of obtained materials was studied using HRTEM (high-resolution transmission electron microscopy) and SEM-EDS (scanning electron microscopy with energy-dispersive X-ray spectroscopy) techniques. TEM (transmission electron microscopy) analyses were carried out to characterize the size and morphology of the particles. Figure 3a,e clearly shows the formation of rectangular prism nanorods with a size being around 40 nm × 20 nm for nHAp: Li^+^ (d) and 37 nm × 17 nm for nHAp: Li^+^/Eu^3+^ (h). The results are in good agreement with the crystallite size estimated based on Debye–Scherrer’s equation (see Table 1). The selected area electron diffraction (SAED) patterns present diffuse spotty rings, indicating the polycrystallinity of nanomaterials (see Figure 3c,g). The ordered rings belong to the lattice plane families consistent with those for bulk hydroxyapatite. The interplanar lattice spacing usually shows a small deviation, typical for nanomaterials with a high surface-to-volume ratio (Figure 3).

The elemental maps were detected, showing the distribution of the ions in the apatite structure (Figure 4b). The content of chemical elements (Ca, P, O, Eu) in the obtained materials was analyzed by the SEM-EDS method (Figure 4c). However, because lithium is too light to be detected by this technique, the ICP-OES method was applied for the effective elemental analysis in the entire sample volume. The amount of Ca, Li, Eu, and P in the material was equal to 358.3 mg/g (RSD = 1.3%), 4.40 mg/g (RSD = 2.1%), 27.06 mg/g (RSD = 1.7%) and 183.5 mg/g (RSD = 2.4%), respectively. The calculated average percentage of Eu^3+^ ions was 1.83 mol%, while Li^+^ ion content was 6.50 mol%. The molar ratio of all cations to phosphorous ions was equal to 1.65 ± 0.02, very close to the stoichiometric n_Ca_/n_P_ (1.67) ratio. All given results were obtained from three independent measurements.

SEM technique was used for surface analysis. The porous structure of the obtained material is clearly visible in Figure 4a. The hydroxyapatites obtained by our group belong to mesoporous materials with pore diameters ranging between 38 nm and 60 nm (depending on the type of dopant), which was described in detail previously [32]. Mesoporous bioactive materials such as nHAp have attracted the attention of many researchers in the field of tissue engineering. They offer chemical and thermal stability, a significantly large surface area within a relatively small volume and, therefore, better surface functionality, higher biocompatibility, and resistance to degradation from the external environment [40]. These features are desirable for effective tissue regeneration. Moreover, mesoporous materials reveal new possibilities for their use in local drug delivery and controlled release of biological agents.

### 2.2. Biological Evaluation

#### 2.2.1. Cell Viability

To assess the cytotoxicity of novel synthetic nano-hydroxyapatite (nHAp), an MTT assay was performed on normal human dermal fibroblast cells (NHDF). No influence of the tested compounds on NHDF cells’ morphology was found for either studied nanohydroxyapatite (Figure 5). After 24 h incubation of NHDF cultures with nHAp: Li^+^, a statistically significant increase in mitochondrial activity at concentrations of 15 and 20 µg/mL compared to the negative control was demonstrated. At the highest tested concentrations, a statistically insignificant decrease in mitochondrial activity was observed. Cell viability was shown to depend on the concentration of the second studied compound (nHAp: Li^+^/Eu^3+^). At the concentration range of 30–50 µg/mL, a statistically significant decrease in mitochondrial activity was demonstrated compared to the negative control. In both nanohydroxyapatites in the tested concentration range, no cytotoxic potential was observed (there was no decrease in the mitochondrial activity below 70% compared to the control).

Similar observations were made by the Yan-Zhong team, who showed that Eu^3+^-doped hydroxyapatite did not exhibit cytotoxic properties against lung cancer cells (A549) [41]. The authors of the study wanted to demonstrate the properties of Eu-doped hydroxyapatite. They are neutral to human cells and could potentially be carriers of cytostatic drugs in cancer treatment. Moreover, it has been shown that such doped HAp can efficiently penetrate the cytoplasm of cells and then bind to DNA without any cytotoxicity [41].

#### 2.2.2. Mitochondrial Activity and Neuronal Features of Cells

Our research model allowed us to check whether the application of tested nHAp influences the regeneration of mechanical damages in neuron-like cells. The study was carried out using differentiated PC12 and SH-SY5Y cell lines. Example micrographs showing cells after differentiation using procedures appropriate for each line are shown in Figure 6.

Mitochondrial activity tests were performed. It is known that as neurite length increases, other mitochondria appear along the length of the neurites, which may influence the higher activity observed in the MTT assay [42].

Neurite damage caused a decrease in mitochondrial activity by approximately 30% in both SH-SY5Y and PC12 cells (Figure 7a,b). The increase in mitochondrial activity was observed compared to the negative control at the concentration range of 2.5–10 µg/mL in SH-SY5Y cell cultures (for nHAp: Li^+^ at concentrations of 2.5 and 5.0 µg/mL, the increase was statistically significant). On the other hand, in PC12 cultures, a statistically significant increase in the number of formazan crystals was observed at concentrations of 2.5–15 µg/mL after incubation with nHAp: Li^+^ and at concentrations of 0.5–15 µg/mL after incubation with nHAp: Li^+^/Eu^3+^. Moreover, cell culture incubation with each tested nanohydroxyapatite resulted in increased mitochondrial activity compared to the positive control in the entire tested concentration range in both SH-SY5Y (statistically significant except for 0.5 µg/mL) and PC12 (statistically significant) cell cultures.

This increase may be related to the length of the neurites. The measured average neurite length in the positive control was approximately 20% shorter than in the negative control (Figure 7c,d). The 24 h incubation with tested nanohydroxyapatite resulted in a statistically significant increase in the neurite length in the entire range of tested concentrations in the SH-SY5Y cultures. Moreover, incubation of SH-SY5Y cultures with nHAp: Li^+^/Eu^3+^ led to the greater average neurite length than in the negative control at the concentration range of 2.5–30 µg/mL (statistically significant for 2.5–20 µg/mL). In the case of PC12 cultures, the increase in the average length of neurites was observed after incubation with nHAp: Li^+^ at the concentration range of 0.5–30 µg/mL (statistically significant at 5–20 µg/mL) and with nHAp: Li^+^/Eu^3+^ in the entire range of studied concentrations (statistically significant at 5–30 µg/mL). Simultaneously, in the concentration range of 5–20 µg/mL of nHAp: Li^+^/Eu^3+^, this increase was statistically significant compared to the negative control.

The correlations between the MTT assay results and the average neurite length were calculated (Table 2). In the case of nHAp: Li^+^, a strong correlation was demonstrated between the mitochondrial activity and the average length of neurites in PC12 cultures and for nHAp: Li^+^/Eu^3+^ in SH-SY5Y cultures. This may suggest the influence of the tested compounds on the alteration in the activity and location of mitochondria along neurites. To confirm this relationship, detailed future studies are planned to elucidate the mechanisms of action of studied nanohydroxyapatites on the neurite’s regeneration. The different effects of correlation may result from the use of two different cell lines of different origins. Even after differentiation, while providing a neurobiological research model, cells can have different properties (e.g., dopamine release). In the next stage, we will determine the cellular mechanisms induced in the presence of nanohydroxyapatite. Simultaneously, to confirm or exclude the influence of the tested compounds on the differentiation and increase in mitochondrial activity, it is planned to test them on primary neuronal cultures.

#### 2.2.3. Reactive Oxygen Species and Nitric Oxide

To assess nanohydroxyapatite’s influence on the reactive oxygen species (ROS) and nitrite ions levels in cultures of neuron-like cells, DCF-DA and Griess assays were performed, respectively. In both cell lines, after the mechanical damage of neurites, a statistically significant increase by several percentage points of ROS and nitric oxide (NO) was observed compared to the negative control (Figure 8). The application of nanohydroxyapatites resulted in the reduction in ROS level similar to that in the negative control (this reduction was statistically significant for nHAp: Li^+^ at the concentration range of 5–10 µg/mL and for nHAp: Li^+^/Eu^3+^ at 0.5–40 µg/mL) in SH-SY5Y cells (Figure 8a). At the same time, the decrease in NO level was observed in SH-SY5Y cultures (statistically significant in the concentration range of 0.5–30 µg/mL for nHAp: Li^+^ and 0.5–20 µg/mL for nHAp: Li^+^/Eu^3+^; Figure 8c). Moreover, the reduction in NO level was statistically significant compared to the negative control in the concentration range of 0.5–30 µg/mL for nHAp: Li^+^ and 15 µg/mL nHAp: Li^+^/Eu^3+^ (Figure 8c).

In PC12 cell cultures, ROS level decrease to a level close to the negative control was also observed in the entire concentration range (Figure 8b). However, this decrease was statistically significant, but only at concentrations of 10–20 µg/mL for nHAp: Li^+^ and 2.5–20 µg/mL nHAp: Li^+^/Eu^3+^. Simultaneously, at moderate concentrations (10–15 µg/mL), a slightly stronger ROS reduction was observed compared to the negative control. In the PC12 cell cultures, a statistically significant decrease in NO level was observed in the concentration range of 0.5–20 µg/mL for both nanohydroxyapatites compared to the positive control (Figure 8d). Moreover, the reduction in NO was also shown compared to the negative control in the concentration range of 5–15 µg/mL (for 10 µg/mL nHAp: Li^+^ statistically significant).

The study showed a reduction in the level of free radicals compared to the control after mechanical damage without the tested hydroxyapatites, which may be due to the influence of Eu^3+^ ions that can reduce the level of free radicals in modeled conditions of inflammation [43]. Based on the common knowledge about the connection between DNA damage and the level of free radicals in cells, we assume that tested nHAp could potentially affect the regeneration of DNA strands [41,43]. In the next stage of the research, it is planned to check whether such regenerative activity occurs.

In this study, we demonstrated the potential properties of nHAp to induce the regeneration of mechanically damaged neuron-like cells. Interestingly, the Sun team [44] showed that nHAp doped with Eu^3+^ could be a good form of gene carriers, e.g., NT-3, to neurons in both in vitro and in vivo models. The results suggest that both nHAp: Li^+^ and nHAp: Li^+^/Eu^3+^ are characterized by unique biological properties, making them suitable for further research as agents activating regenerative processes in cells, including neurons.

## 3. Materials and Methods

### 3.1. Physicochemical Analysis

#### 3.1.1. Preparation of nHAp: Li^+^ and nHAp: Li^+^/Eu^3+^

A series of Ca_10_(PO_4_)_6_(OH)_2_ powders doped with Li^+^ and co-doped with Li^+^/Eu^3+^ were synthesized by a wet chemistry method. Ca(NO_3_)_2_·4H_2_O (≥99% Acros Organics, Geel, Belgium), (NH_4_)_2_HPO_4_ (≥98% Avantor Performance Materials Poland S.A, Gliwice, Poland), Eu_2_O_3_ (99.99% Alfa Aesar, Ward Hill, MA, USA), Li_2_CO_3_ (99% Alfa Aesar, Ward Hill, MA, USA) were used as the starting reagents. The pH was regulated by NH_3_·H_2_O (99% Avantor Performance Materials Poland S.A, Gliwice, Poland). The obtained materials were annealed at 500 °C in order to remove undesirable by-products and for better crystallization of the final product.

Firstly, appropriate amounts of Li_2_CO_3_ and Eu_2_O_3_ were digested in excess HNO_3_ (65%; Suprapur Sigma-Aldrich, St. Louis, MO, USA) in order to transform them into the water-soluble lithium and europium nitrates, respectively, and then they were re-crystallized three times. Afterwards, Ca(NO_3_)_2_·4H_2_O was dissolved in 50 mL of MQ-water together with LiNO_3_ and Eu(NO_3_)_3_. The total molar content of the cations was set to 10 mmol. Subsequently, 30 mL of (NH_4_)_2_HPO_4_ (6 mmol) aqueous solution was added to the mixture. A white precipitate was formed. The suspension pH was adjusted to 10 with NH_3_·H_2_O under constant stirring at 100 °C for 2 h. Finally, the precipitate was dried for 24 h at 90 °C and thermally treated at 500 °C for 3 h, resulting in the formation of fine-grained powders. The same procedure was used for the final product containing only Li^+^ (un-doped with Eu^3+^ ions).

#### 3.1.2. Apparatus and Analysis Methods

PANalytical X’Pert Pro X-ray diffractometer (Malvern Panalytical Ltd., Malvern, UK) with a Cu–Kα radiation over a 2θ range from 10° to 60° was used to determine phase purity and crystallite size. Match! and Origin softwares were employed to analyze the X-ray diffraction (XRD) profiles. The obtained patterns were compared to standards gathered from the Inorganic Crystal Structure Database (ICSD–26204). The average crystallite sizes (*D*) of both materials was estimated from the full-width at half-maximum (*β*) of the diffraction peaks, using Debye–Scherrer’s method:(1)$$D=\frac{k\lambda }{\beta cos\theta }$$
where *λ* is the wavelength of the incident X-ray radiation (0.154 nm), *k* is a constant, and *θ* is the Bragg’s diffraction angle.

The lattice parameters (*a* = *b* and *c*) were evaluated based on Miller indices (*hkl*) and Bragg’s law:(2)nλ=2dsinθ
where *n* is a positive integer, *λ* is the wavelength of the incident wave, and *d* is the interplanar spacing.

The crystallinity degree as a fraction of crystalline phase presents in the examined volume was estimated using the formula [45]:(3)$$X_{c}\approx 1-\left(V_{112/300}/I_{300}\right)$$
where *I*_300_ is the intensity of (300) reflection, and *V*_112/300_ is the intensity of the valley between (112) and (300) reflections.

In-plane (*ε_a_*) and out-of-plane (*ε_c_*) strains were estimated by comparing the experimental results of lattice parameters (*a* and *c*) with the reference values (*a*_0_ and *c*_0_), using the following equations [46]:(4)$$\varepsilon_{a}=\frac{a-a_{0}}{a_{0}}\;$$
(5)$$\varepsilon_{c}=\frac{c-c_{0}}{c_{0}}\;$$

Fourier-transform infrared spectroscopy (FTIR, Thermo Fisher Scientific Nicolet iS50 FT-IR spectrometer, Waltham, MA, USA) in a frequency range of 4000−400 cm^−1^ with a 0.5 cm^−1^ scan step was applied to evaluate the functional groups of the obtained materials. The number of replicates was 5. The size of the particles and morphology of obtained materials were observed using a high-resolution transmission electron microscopy (HRTEM, Philips CM-20 Super Twin microscope, operating at 200 kV with a resolution of 0.24 nm, Eindhoven, The Netherlands). Element analysis was performed using a scanning electron microscope (SEM, FEI Nova NanoSEM 230, Hillsboro, OR, USA) with an energy-dispersive X-ray spectrometer (EDS, Genesis XM4) in order to confirm the elemental composition and identify the spatial distribution of the Ca and P, O and Eu. Appropriate concentrations of the elements were determined from the calibration curve by using ICP-OES Agilent 720 apparatus (Agilent Technologies, Inc., Santa Clara, CA, USA).

### 3.2. Cell Culture Studies

#### 3.2.1. Cell Lines

The current study was carried out using SH-SY5Y, PC12 and NHDF cell lines purchased from the ATCC (Manassas, VA, USA). The SH-SY5Y cells were differentiated with retinoic acid, and PC12 cells with NGF towards neuron-like cells. Cells were cultured in a humidified environment at 37 °C with 5% CO_2_ and passaged twice a week by transferring to the tube and centrifuging at 1000× *g* for 5 min. Then, the supernatants were removed, and cells were resuspended in fresh media. In the case of the PC12 cell line, cell clumps were broken by twofold squeezing through a 0.7 mm needle. In biological studies, PC12 cells were used at passages 11–16 and SH-SY5Y cell line was used at passages 9–14.

#### 3.2.2. Cell Culture Media

For SH-SY5Y and PC12 cells, two types of cell culture medium were used in the present study: medium to grow cells (primary medium) and medium to differentiate cells (differentiation medium), while for NHDF cells, only one (primary) medium was used.

The SH-SY5Y cells were cultivated in MEM supplemented with 10% FBS, 2 mM l-glutamine, 1.25 µg/mL amphotericin B and 100 µg/mL gentamicin. Medium for SH-SY5Y cell differentiation was based on MEM with l-glutamine, amphotericin B, and gentamicin with 1% FBS, and with the addition of 10 µM retinoic acid. The PC12 cells were grown in medium containing RPMI-1640 supplemented with 10% donor horse serum (DHS), 5% fetal bovine serum (FBS), 2 mM l-glutamine, 1.25 µg/mL amphotericin B and 100 µg/mL gentamicin. These cells were differentiated in the medium which also consisted of RPMI-1640, l-glutamine, amphotericin B and gentamicin but without FBS, only with 1% DHS, and with the addition of 100 ng/mL NGF. The NHDF cells were cultured in DMEM without phenol red and supplemented with 10% FBS, 2 mM l-glutamine, 1.25 µg/mL amphotericin B and 100 µg/mL gentamicin. All prepared cell culture media were kept at 4–8 °C for up to one month.

#### 3.2.3. Tested Compounds

The tested novel synthetic nanohydroxyapatites (nHAp: Li^+^ and nHAp: Li^+^/Eu^3+^) were dissolved in DMSO. All prepared stock solutions were kept at −20 °C for up to 6 months. To achieve the final concentration, the compounds were dissolved in the primary medium, which was also used in the assays.

#### 3.2.4. Modification of Cell Culture Plates Surface

The surface of wells of cell culture plates was modified only for PC12 cells to enable adhesion to the plastic surface. To this end, type I collagen (Sigma Aldrich, Steinheim, Germany) was dissolved in 0.1 M acetic acid to the concentration of 0.1% (*w*/*v*). Such prepared collagen stock solution was kept at −20 °C for up to 6 months. After diluting the stock solution in distilled water to the concentration of 0.01% (*w*/*v*), this final solution was added to the wells in a volume needed to coat their surface. Afterwards, the plates were maintained at 4–8 °C overnight. After this time, the residual collagen solution was removed, and the plates were washed three times with PBS. Collagen-coated plates were kept at 4 °C for up to one month. Prior to usage, collagen-coated plates were irradiated with UV for 30 min.

#### 3.2.5. Experimental Design

Cells were seeded in multi-well cell culture plates at a density of 10,000 cells/well. In cultures of SH-SY5Y and PC12 cells, 24 h after seeding, the primary medium was removed. An appropriate differentiation medium was then added to induce neuronal differentiation— with retinoic acid for 5 days in SH-SY5Y cultures, or with NGF for 3 days in PC12 cells.

In order to evaluate the impact of the studied compounds on cellular viability, the MTT assay was performed on the NHDF cells according to ISO 10993 standard part V Annex C. The SH-SY5Y and PC12 cells were then mechanically damaged, and novel nano-hydroxyapatites were added to evaluate the functioning of the mitochondria during the MTT assay and assess whether these new compounds can exert neuroregenerative activity on mechanically damaged cells. To evaluate neuronal features of cells, the average neurite length was measured spectrofluorimetrically. Next, the DCF-DA and Griess assays were carried out to determine the tested compounds’ impact on the reactive oxygen species and nitrite ion levels.

All experiments were performed in 96-well culture plates. The cell density in the MTT assay was 10,000 per well. For the DCF-DA assay, cells were plated at a density of 25,000 per well. In contrast, to assess the neurite length, 5000 cells per well were seeded so that the neurite length could be accurately measured.

Mechanical cell damage was induced by making two scratches intersecting at right angles in the center of the well. In order for the damage to be reproducible (with the same force), it was performed with a multichannel pipette. In the current study, two controls were used: the negative control (as a reference), which were cells incubated in the primary medium without mechanical damage and without tested compounds; and the positive control, which were cells subjected to mechanical damage but without tested compounds.

#### 3.2.6. MTT Assay

The viability assay was carried out as described in the paper by Grajzer et al. [47]. After 24 h incubation of NHDF cells with studied compounds, the supernatant was removed, cell culture was washed, 1 mg/mL MTT solution in MEM (without supplements) was added to each well, and plates were incubated for 2 h at 37 °C. After incubation, the medium was removed, and formazan crystals were dissolved in 100 µL of isopropanol for 30 min. Absorbance was measured at 570 nm with a Varioskan Multimode Microplate Reader (Thermo Fisher Scientific, Waltham, MA, USA).

#### 3.2.7. Length of Neurites

After 24 h incubation of SH-SY5Y and PC12 cells with studied compounds, the average length of their neurites was measured for 50 cells in each of 5 independent replicates, relying on microscopic images using the ImageJ software. Only cells that had neurite with length twice as long as the cell body diameter were assessed.

#### 3.2.8. DCF-DA Assay

The intracellular level of ROS in SH-SY5Y and PC12 cell lines was measured by the DCF-DA assay. After 1 h incubation with studied compounds, the culture medium was removed, and cells were washed with PBS. Then, 25 µM DCF-DA solution in an appropriate medium without serum and phenol red was added for 1 h at 37 °C. The ROS level was evaluated fluorimetrically using a Varioskan Multimode Microplate Reader (Thermo Fisher Scientific, Waltham, MA, USA) with excitation at 485 nm and emission at 535 nm.

#### 3.2.9. Griess Assay

Nitrite ion levels in SH-SY5Y and PC12 cells were evaluated by Griess assay. After 1 h incubation of cells with studied compounds, 50 µl of supernatant was transferred into a new plate, and Griess reagents—50 µL of 0.1% *N*-(1-naphthyl)ethylenediamine dihydrochloride and 50 µL of 1% sulfanilamide in 5% phosphoric acid were added. Such prepared plates were left for 20 min in the dark at room temperature. Nitrite ion levels were measured at a wavelength of 548 nm using the Varioskan Multimode Microplate Reader (Thermo Fisher Scientific, Waltham, MA, USA).

#### 3.2.10. Statistical Analysis

All biological studies were performed in five independent experiments. Each experiment was conducted in six replications. In biological assays, two controls were used. One was a culture without compound and without mechanical damage (the negative control). The second was the culture without tested compounds but subjected to mechanical damage (the positive control). All data are presented as mean values with a standard error of the mean (SEM) and expressed as E/E_0_ ratios, where E is the negative control. The E/E_0_ ratio was also calculated for the culture without tested compounds but subjected to mechanical damage (the positive control). Therefore, the presented graphs show the activity of studied nanohydroxyapatites (nHAp: Li^+^ and nHAp: Li^+^/Eu^3+^) compared to the cultures of healthy cells as well as mechanical damage subjected cultures. Statistical significance was analyzed using one-way analysis of variance (ANOVA) and multiple comparisons with Tukey’s post hoc test. Pearson correlation coefficients were determined to evaluate the relationship between studied parameters. Statistical analyses were performed with Statistica 13.0 software (Dell Software Inc., Round Rock, TX, USA) with *p* < 0.05, *p* < 0.01 and *p* < 0.001 adopted as the significance levels.

## 4. Conclusions

The proposed nanocrystalline apatites doped and co-doped with Li^+^ and Eu^3+^ ions may be a very attractive biomaterial for the regeneration of nervous tissue. The physicochemical characterization indicated that the obtained apatites crystallize in a hexagonal structure, belong to mesoporous materials with high specific surface area, and the particles possess nanorod morphology. The lithium acts as a hydroxyapatite crystallization enhancer, while europium ions introduce additional properties to the apatite—the material can be monitored in real time based on a phenomenon called photoluminescence. Moreover, lithium shows great potential for nerve regeneration. In this work, we showed that Li^+^-doped nHAp affects the increase in mitochondrial activity and the length of neurites in cells in which mechanical damage was previously induced. At the same time, a decrease in ROS and NO levels was also observed. We found that europium ions influence neuronal features even more strongly than doping with lithium alone. Preliminary biological studies of nanohydroxyapatites were performed on continuous cell lines. In order to confirm their biological potential, tests should now be carried out on the primary neuronal cultures.

## Figures and Tables

**Figure 1 ijms-22-04454-f001:**
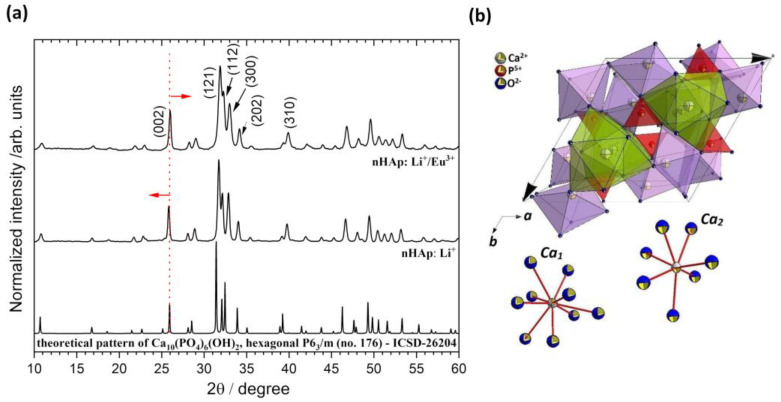
X-ray diffraction patterns of the Ca_10_(PO_4_)_6_(OH)_2_ (nHAp) doped and co-doped with Li^+^ (nHAp: Li^+^) and Eu^3+^ (nHAp: Li^+^/Eu^3+^) and annealed at 500 °C with the indication of lattice planes (**a**). Projection of the hydroxyapatite unit cell with the coordination of Ca_1_ and Ca_2_ crystallographic sites (**b**).

**Figure 2 ijms-22-04454-f002:**
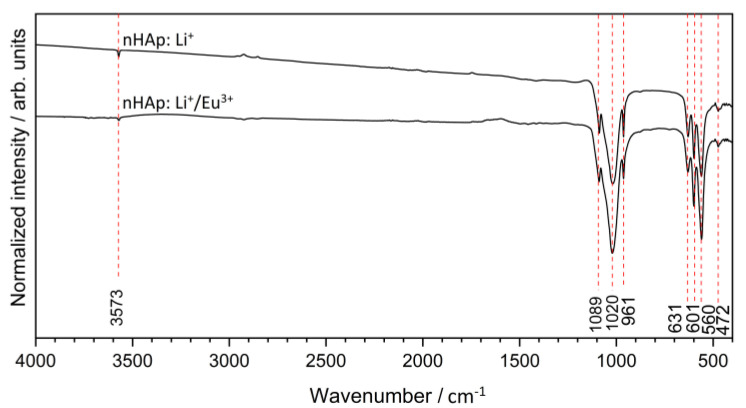
FT-IR spectra of the Ca_10_(PO_4_)_6_(OH)_2_ (nHAp) doped and co-doped with Li^+^ (nHAp: Li^+^) and Eu^3+^ (nHAp: Li^+^/Eu^3+^), and annealed at 500 °C.

**Figure 3 ijms-22-04454-f003:**
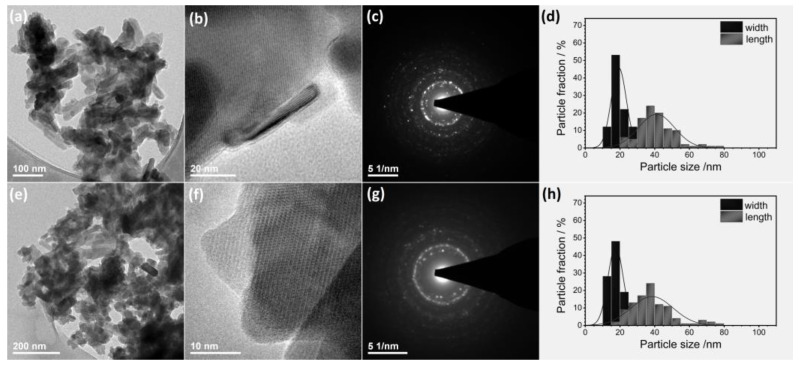
TEM and SAED images of the nHAp: Li^+^ (**a**–**c**) and nHAp: Li^+^/Eu^3+^ (**e**–**g**). Histograms of the particle size distribution (lengthwise and widthwise diameters) for nHAp: Li^+^ (**d**) and nHAp: Li^+^/Eu^3+^ (**h**).

**Figure 4 ijms-22-04454-f004:**
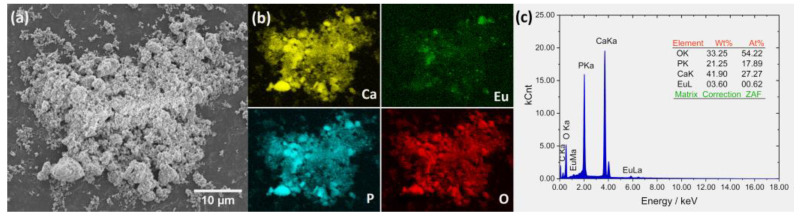
SEM image (**a**) with EDS elemental distribution maps (**b**) and EDS spectra (**c**) for nHAp: Li^+^/Eu^3+^.

**Figure 5 ijms-22-04454-f005:**
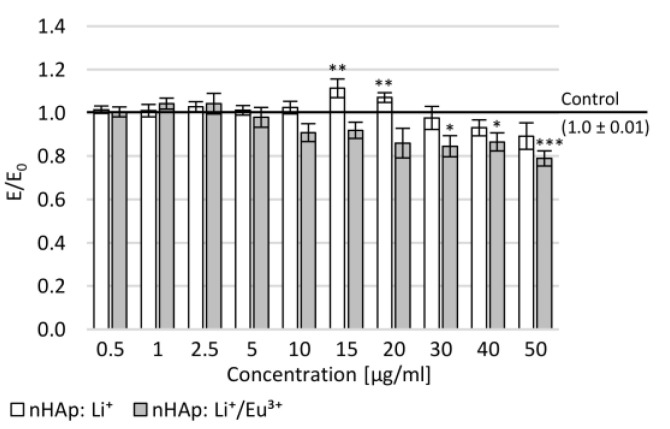
Effect of tested nanohydroxyapatites (nHAp: Li^+^ and nHAp: Li^+^/Eu^3+^) on the metabolic activity of NHDF cells measured in the MTT assay; Control (the negative control)—cell culture incubated without tested substances; * *p* < 0.05, ** *p* < 0.01, *** *p* < 0.001—significant difference compared to the negative control.

**Figure 6 ijms-22-04454-f006:**
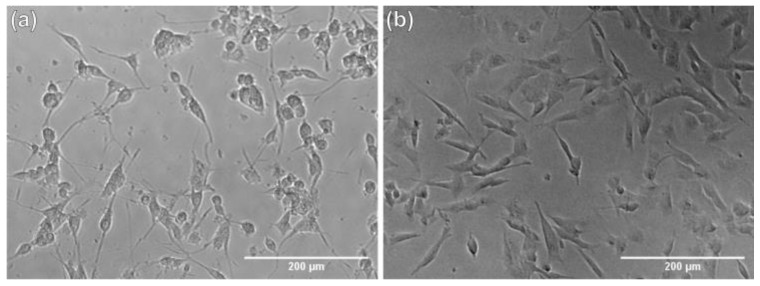
Example micrographs showing differentiated PC12 cells (**a**) and SH-SY5Y cells (**b**).

**Figure 7 ijms-22-04454-f007:**
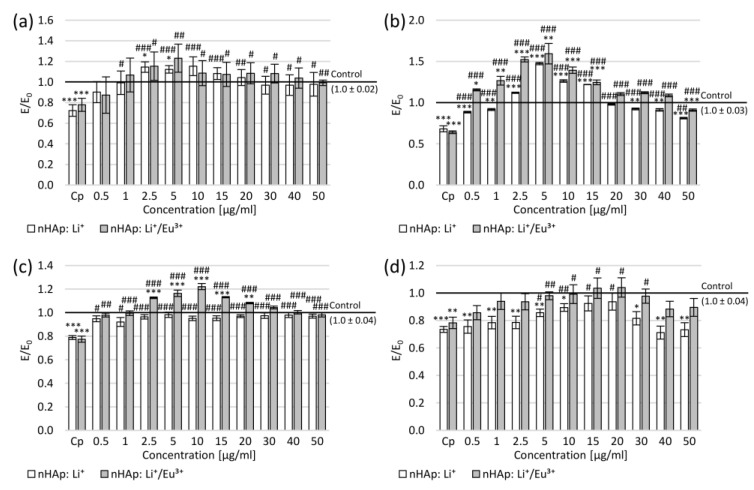
Effect of tested nanohydroxyapatites (nHAp: Li^+^ and nHAp: Li^+^/Eu^3+^) on the SH-SY5Y (**a**,**c**) and PC12 (**b**,**d**) cells after mechanical damage: (**a**,**b**) metabolic activity measured in the MTT assay; (**c**,**d**) average length of neurites (data shown as ratio of the average length of neurites in treated cells compared to control cells); Control (the negative control)—cell culture incubated without nanohydroxyapatites and without mechanical damage; Cp (the positive control)—cell culture after mechanical damage but without tested compounds; * *p* < 0.05, ** *p* < 0.01, *** *p* < 0.001—significant difference compared to the negative control; # *p* < 0.05, ## *p* < 0.01, ### *p* < 0.001—significant difference compared to the positive control.

**Figure 8 ijms-22-04454-f008:**
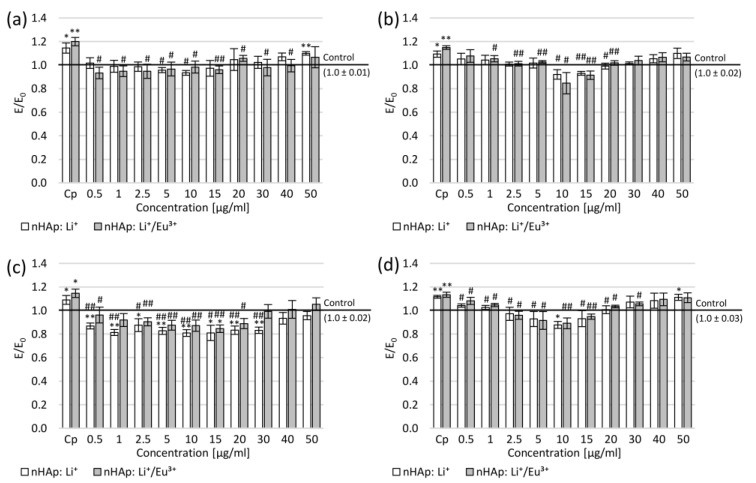
Effect of tested nanohydroxyapatites (nHAp: Li^+^ and nHAp: Li^+^/Eu^3+^) on the SH-SY5Y (**a**,**c**) and PC12 (**b**,**d**) cells after mechanical damage: (**a**,**b**) DCF-DA assay; (**c**,**d**) Griess assay; Control (the negative control)—cell culture incubated without nanohydroxyapatites and without mechanical damage; Cp (the positive control)—cell culture after mechanical damage but without tested compounds; * *p* < 0.05, ** *p* < 0.01—significant difference compared to the negative control; # *p* < 0.05, ## *p* < 0.01—significant difference compared to the positive control.

**Table 1 ijms-22-04454-t001:** Unit cell parameters (a, c), cell volume (V), average crystallites size (D), crystallinity degree (χ_c_), in-plane (ε_a_) and out-of-plane (ε_c_) strains, for the Ca_10_(PO_4_)_6_(OH)_2_ (nHAp) doped and co-doped with Li^+^ (nHAp: Li^+^) and Eu^3+^ (nHAp: Li^+^/Eu^3+^), and annealed at 500 °C.

Sample	a (Å)	c (Å)	V (Å^3^)	D (nm)	Χ_c_ (%)	|ε_a_| (%)	|ε_c_| (%)
Ca_10_(PO_4_)_6_(OH)_2_, single crystal [38]	9.424(4)	6.879(4)	529.09(4)	-	-	-	-
nHAp: Li^+^	9.427(7)	6.892(7)	530.55(2)	46.63	73.9	0.035	0.193
nHAp: Li^+^/Eu^3+^	9.392(0)	6.852(8)	523.49(8)	38.99	61.0	0.344	0.387

**Table 2 ijms-22-04454-t002:** Pearson’s correlation coefficients between the mitochondrial activity and the average length of neurites for the Ca_10_(PO_4_)_6_(OH)_2_ (nHAp) doped and co-doped with Li^+^ (nHAp: Li^+^) and Eu^3+^ (nHAp: Li^+^/Eu^3+^), and annealed at 500 °C.

Cells	Compounds	Length of Neurites vs. Mitochondrial Activity
PC12	nHAp: Li^+^	0.59
	nHAp: Li^+^/Eu^3+^	0.28
SH-SY5Y	nHAp: Li^+^	0.15
	nHAp: Li^+^/Eu^3+^	0.68

## Data Availability

The data presented in this study are available on request from the corresponding author.

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
