# Peer review of "Nanohydroxyapatite as a Biomaterial for Peripheral Nerve Regeneration after Mechanical Damage—In Vitro Study"

_ijms, 2021, doi:10.3390/ijms22094454_

Round 1

Reviewer 1 Report

The paper discusses the use of ion-doped hydroxyapatite (obtained by wet synthesis) for nerve regeneration.

The rationale for using Li- and Eu-doped HA shall be briefly indicated also in the abstract. Also, the abstract is completely focused on nerves, while a great part of the introduction is focused on bone regeneration and the use of HA for this scope. This is misleading.

The english form of the introduction is poor and shall be revised.

The author state that "There is hydroxyapatite (HAp) in the body with a 47 Ca/P molar ratio of at least 1.67, crystallizing in a hexagonal structure [5].". Please note that stoichiometric hydroxyapatite is never present in biological tissues, that instead are formed by biogenic hydroxyapatite, which is nanocrystalline and multi-doped.

The Authors state that "Due to the similarity to the bone, no cytotoxic or 50 carcinogenic effects are observed after implantation of synthetic hydroxyapatite. ". Please note that there is no reason why a material dissimilar from bone should be carcinogenic.

In the introduction, discussion over HA, ion-doped HA and its application is oversimplified and scarcely correct. More in detail:

  • Lines 53-61. The discussion over porous/dense hydroxyapatite is poor, as in fact HA is used in several forms and not just this two (i.e. nanoparticles, nanocoatings, etc). In addition, solubility of the compound does not only (nor even mainly) depend on porosity, but also on crystallinity, presence/absence of ion-doping causing lattice distortion, surface morphology and topography, roughness etc.
  • Lines 62-84. This part is very confused, too simplified and in some parts incorrect.

The rationale for proposing hydroxyapatite for the nerves shall be clarified, as well as that for using europium.

In the results, differences (including broadness of the HA bands and additional bands in the 2teta 21-23 and 28-30° areas) indicate differences in crystallinity degree and also in phase formation in the different specimens. This shall be better commented.

In the FT-IR, the main band for phosphate stretching in HA is at 1030 and not 1020. Bands at 1020 indicate presence of octacalcium phosphate or carbonated hydroxyapatite. A curve of stoichiometric HA must be added to the graph.

The Authors comment that "Both spectra are very similar to each other, and only slight differences are observed in their maximum values depending on the type of dopants (Eu3+ and/or Li+ ) and, therefore, the character of the M‒O interaction ". It is unclear which changes they refer to. This shall be commented.

Please, provide more SEM images, to better highlight if the morphology is homogeneous or not, and if only some areas are porous (as it seems now in Figure 4).

When commenting upon cytotoxicity on fibroblasts, the Authors state that: "Similar observations were made by the Yan-Zhong team, who showed that Eu3+ doped hydroxyapatite did not exhibit cytotoxic properties against lung cancer cells (A549) 223 [33]. The authors of the study wanted to demonstrate the properties of Eu-doped hydroxyapatite. They are neutral to human cells and could potentially be carriers of cytostatic drugs in cancer treatment. Moreover, it has been shown that such doped HAp can efficiently penetrate the cytoplasm of cells and then bind to DNA without any cytotoxicity [33]." However, toxicity against lung cancer has nothing to deal with that on fibroblasts, so the comment is not pertinent.

Methods.

Please indicate the pH used for HA-forming solutions.

Indicate resolution, scan step size and accumulations for FT-IR analyses.

Indicate the number of replicates used for each test. For biological test, please indicate the number of technical and biological replicates. In absence of these data, all results have no significance.

For biological tests, please better define what you use as control. Non doped HA shall also be analyzed in each test.

Conclusions are not supported by the data. Results are not sufficient to demonstrate that lithium can induce nerve regeneration. In addition, lithium reduces HA crystallinity (see XRD graphs), while the Authors state the opposite.

Author Response

Dear Editor,

We would like to express our sincerest gratitude to the Reviewer for the enormous efforts in criticizing the manuscript. We have considered all raised question here follows the detailed answers to the Reviewer. All changes we have made to the original manuscript are marked in the red colour in the text.

Reviewer 1

Comments and Suggestions for Authors

The paper discusses the use of ion-doped hydroxyapatite (obtained by wet synthesis) for nerve regeneration. The rationale for using Li- and Eu-doped HA shall be briefly indicated also in the abstract.

Q1:  Also, the abstract is completely focused on nerves, while a great part of the introduction is focused on bone regeneration and the use of HA for this scope. This is misleading.

Answer: The introduction section has been clarified.

Q2:  The English form of the introduction is poor and shall be revised.

Answer: The introduction section has been revised.

Q3:  The author state that "There is hydroxyapatite (HAp) in the body with a 47 Ca/P molar ratio of at least 1.67, crystallizing in a hexagonal structure [5].". Please note that stoichiometric hydroxyapatite is never present in biological tissues, that instead are formed by biogenic hydroxyapatite, which is nanocrystalline and multi-doped.

Answer: We agree with this comment. The information about hydroxyapatite in the body was clarified.

Q4:  The Authors state that "Due to the similarity to the bone, no cytotoxic or 50 carcinogenic effects are observed after implantation of synthetic hydroxyapatite. ". Please note that there is no reason why a material dissimilar from bone should be carcinogenic.

Answer: We agree with this comment, thus the statement was changed.

Q5:  In the introduction, discussion over HA, ion-doped HA and its application is oversimplified and scarcely correct. More in detail:

  • Lines 53-61. The discussion over porous/dense hydroxyapatite is poor, as in fact HA is used in several forms and not just this two (i.e. nanoparticles, nanocoatings, etc). In addition, solubility of the compound does not only (nor even mainly) depend on porosity, but also on crystallinity, presence/absence of ion-doping causing lattice distortion, surface morphology and topography, roughness etc.
  • Lines 62-84. This part is very confused, too simplified and in some parts incorrect.

Answer: The discussion in the introduction section was improved.

Q6:  The rationale for proposing hydroxyapatite for the nerves shall be clarified, as well as that for using europium.

Answer: It was clarified in the introduction section.

Q7:  In the results, differences (including broadness of the HA bands and additional bands in the 2teta 21-23 and 28-30° areas) indicate differences in crystallinity degree and also in phase formation in the different specimens. This shall be better commented.

Answer: The data of the XRD measurements was processed once again and corrected. The discussion has been improved in the manuscript.

Q8:  In the FT-IR, the main band for phosphate stretching in HA is at 1030 and not 1020. Bands at 1020 indicate presence of octacalcium phosphate or carbonated hydroxyapatite. A curve of stoichiometric HA must be added to the graph.

Answer:  The FT-IR spectrum of the stoichiometric nHAp was not added to the graph, because it was not the subject of presented research. In turn, we have referred it in our earlier work, where the band at about 1020 cm-2 was detected [Marycz, K.; Sobierajska, P.; Smieszek, A.; Maredziak, M.; Wiglusz, K.; Wiglusz, R.J. Li+ activated nanohydroxyapatite doped with Eu3+ ions enhances proliferative activity and viability of human stem progenitor cells of adipose tissue and olfactory ensheathing cells. Further perspective of nHAP:Li+, Eu3+ application in theranostics. Mater. Sci. Eng. C 2017, 78, 151–162, doi:10.1016/j.msec.2017.04.041 ].

Q9:  The Authors comment that "Both spectra are very similar to each other, and only slight differences are observed in their maximum values depending on the type of dopants (Eu3+ and/or Li+) and, therefore, the character of the M‒O interaction ". It is unclear which changes they refer to. This shall be commented.

Answer:  Thank you for this comment. After additional analysis of presented spectra, we did not see significant changes in FT-IR spectra and therefore this information was excluded from the manuscript.

Q10:  Please, provide more SEM images, to better highlight if the morphology is homogeneous or not, and if only some areas are porous (as it seems now in Figure 4).

Answer:  The SEM image has been corrected to be better highlighted regarding its porous form (see Figure 4). The particle morphology has been shown in Figure 3. The particles are elongated in one direction and form nanorods with a tendency to agglomerate.

Q11:  When commenting upon cytotoxicity on fibroblasts, the Authors state that: "Similar observations were made by the Yan-Zhong team, who showed that Eu3+ doped hydroxyapatite did not exhibit cytotoxic properties against lung cancer cells (A549) 223 [33]. The authors of the study wanted to demonstrate the properties of Eu-doped hydroxyapatite. They are neutral to human cells and could potentially be carriers of cytostatic drugs in cancer treatment. Moreover, it has been shown that such doped HAp can efficiently penetrate the cytoplasm of cells and then bind to DNA without any cytotoxicity [33]." However, toxicity against lung cancer has nothing to deal with that on fibroblasts, so the comment is not pertinent.

Answer:   It has been demonstrated with this information that doping with Eu3+ ions does not even have any cytotoxic potential on tumour lines (where such an effect would be expected). To our best knowledge, there is no work confirming the influence of Eu3+ ions on normal cells.

Q12:  Methods. Please indicate the pH used for HA-forming solutions. Indicate resolution, scan step size and accumulations for FT-IR analyses. Indicate the number of replicates used for each test.

Answer:  The information has been added. The pH value during the synthesis was 10, as was previously reported in Materials and Method section.

Q13:  For biological test, please indicate the number of technical and biological replicates. In absence of these data, all results have no significance.

Answer:   All biological studies were performed in five independent experiments. Each experiment was done in six replications. We have supplemented this information in the Materials and Methods section.

Q14:  For biological tests, please better define what you use as control. Non doped HA shall also be analyzed in each test.

Answer:  In biological assays, two control was used. One was a culture without compound and without mechanical damage (the negative control). The second was the culture without tested compounds but subjected to mechanical damage (the positive control). The study aimed to evaluate the effect of nHAp doped with Eu3+ and Li+ ions and not with hydroxyapatite itself. Hydroxyapatite was previously tested and found to be non-toxic (quote). As we have completed the information in the introduction, it is well known that nHAp has a neuroregenerative effect. However, our work aimed to check the influence of admixtures, not of nHAp itself.

Q15:  Conclusions are not supported by the data. Results are not sufficient to demonstrate that lithium can induce nerve regeneration. In addition, lithium reduces HA crystallinity (see XRD graphs), while the Authors state the opposite.

Answer:  Comparing the neurite lengths after applying the tested substances, longer neurites were observed than without control in the medium only. Based on XRD data and results gathered in table 1, it is clearly seen that lithium increases HAp crystallinity. The crystallinity rate for nHAp: Li+ was about 74%, while for nHAp:Li+/Eu3+ this value was lower (61%). So, the europium ions reduce nHAp crystallinity in this case.

Reviewer 2 Report

It is a nice research work, I would like to ask the authors

1. to specify more precise the preparation HAp, because we know that also the concentration of raw materials are important for the product properties like particle size, grade of crystallinity etc... 

2. The question is, how the authors checked the Ca/P rate of the product samples? Only by EDS? Figure 4 show us that the Ca/P rate=1,53.

3.

"For this sample, the reduction in FWHM (full width at half maximum) of the XRD diffraction peaks indicates that lithium acts as a hydroxyapatite crystallization enhancer.

In this work, we showed that Li+-doped nHAp affects the increase in mitochondrial activity and the length of neurites in cells in which mechanical damage was previously induced."

Please explain more detailed how the crystallisation grade of nHap influences the biological properties! Why Li+ could have benefit role for these properties, as enhancer of crystallisation of HAp?

Author Response

Dear Editor,

We would like to express our sincerest gratitude to the Reviewer for the enormous efforts in criticizing the manuscript. We have considered all raised question here follows the detailed answers to the Reviewer. All changes we have made to the original manuscript are marked in the red color in the text.

Reviewer 2

Comments and Suggestions for Authors

It is a nice research work, I would like to ask the authors:

Q1:  To specify more precise the preparation HAp, because we know that also the concentration of raw materials are important for the product properties like particle size, grade of crystallinity etc... 

Answer: The preparation section was improved.

Q2: The question is, how the authors checked the Ca/P rate of the product samples? Only by EDS? Figure 4 show us that the Ca/P rate=1,53.

Answer: The Ca/P rate equal to 1.53 only when we do not take into account the concentration of Li+ ions. For this reason, we attached to the manuscript the results of ICP-OES analysis, where the content of all cations and phosphorus ions were determined. The calculated (Ca+Eu+Li)/P ratio was equal to 1.65 ± 0.02, very close to the stoichiometric nCa/nP (1.67) value.

Q3: "For this sample, the reduction in FWHM (full width at half maximum) of the XRD diffraction peaks indicates that lithium acts as a hydroxyapatite crystallization enhancer.

In this work, we showed that Li+-doped nHAp affects the increase in mitochondrial activity and the length of neurites in cells in which mechanical damage was previously induced."

Please explain more detailed how the crystallisation grade of nHap influences the biological properties! Why Li+ could have benefit role for these properties, as enhancer of crystallisation of HAp?

Answer: Thank you for this comment. The above information has been explained detailed in the introduction section: From a medical point of view, poorly crystalline or even amorphous hydroxyapatite is unfavorable for bone grafting due to its very fast dissolution rate. Many studies found out that an ideal material for orthopedics was HAp with a Ca/P molar ratio close to that of stoichiometric value equal to 1.67 [3,6–8]. It was reported that required resorbability of hydroxyapatite can be achieved by controlling its degree of crystallinity to 60–70% and reducing its particle size to the nanolevel [9]. In our work, lithium ions enhance the crystallization grade up to 74%. On the other hand, for nHAp:Li+/Eu3+ this degree equal to 61% is also sufficient for biological properties.

This manuscript is a resubmission of an earlier submission. The following is a list of the peer review reports and author responses from that submission.

Round 1

Reviewer 1 Report

The start of this paper, finding new biomaterials is a good thing. I also think that there may be some good things in here, but the paper really needs a through revision. When that is said I have several problems with the paper, that I think that are so severe that I am not sure if the results have any meaning.

  1. The over all English is very bad and need new much work
  2. The way the references are set up must be altered. many of the referrals to authors lack proper references. 
  3. When referring to the figures, it must be the correct figure. This is not the case throughout the paper
  4. The figures are difficult to understand and in many of the experiments, the necessary controls are lacking.
  5. The methods described are not well characterized. How are the cells damaged for example?
  6. When testing new biomaterials for peripheral nerve regeneration, it is often tested on primary rat or mice DRG neurons. In this paper cell lines have instead been used. One cell line PC12, that originate from adrenal medulla and bone marrow and SH-SY5Y  from biopsy taken from a four-year-old female with neuroblastoma. These cells do not have a PNS origin and one is of human and one is from a rat origin. Both are cell lines (cancer back ground) and of course regenerate readily. The cell lines are also treated differently and grown on different cell culture surfaces.
  7. When differentiating cell lines it should be obvious for the reader than they actually turn into neuron like cells, but in this case no such figures or picture of the cells are present.
  8. The entire disposition of the paper needs to be changed. Parts of the results are not results, but more an insurance why this paper is so important! 
  9. Then comes cell culture: Everyone that has grown these two cell lines know that this is not difficult. A bit of antibiotics in the culture is ok, but fungicides? The addition that the slides coated with fibronectin need UV light after coating to presumably kill bacteria? This group really need to work on their sterile technique! Also fungicides often impede cell growth.
  10. It is also well known that the two cell lines cannot be passaged for ever. In fact with each passage the proliferation decrease. The paper does not state what the passage was when they started their experiment and when they stopped. In the case of SH-SY5Y cells we have seen a significant decrease in proliferation after 20 passages fx.
  11. The authors are not able to convince me that their biomaterial is really good. How would it work in a nerve repair setting? Would something that is har as enamel really be good? How would it be degraded in the body?

Author Response

Dear Editor,

We would like to express our sincerest gratitude to the Reviewer for the enormous efforts in criticizing the manuscript. We have considered all raised question here follows the detailed answers to the Reviewer. All changes we have made to the original manuscript are marked in the red color in the text.

Reviewer 1

The start of this paper, finding new biomaterials is a good thing. I also think that there may be some good things in here, but the paper really needs a through revision. When that is said I have several problems with the paper, that I think that are so severe that I am not sure if the results have any meaning.

Q1:  The over all English is very bad and need new much work.

Answer:

The manuscript has been improved by an English lecturer and also checked with automated grammar checking software (according to American English). The paper contained linguistic errors, but we do not agree that they deserved the description “all English is very bad”.

Q2:  The way the references are set up must be altered. many of the referrals to authors lack proper references. 

Answer:

The references have been checked.

Q3: When referring to the figures, it must be the correct figure. This is not the case throughout the paper

Answer:

We have corrected references to the figures throughout the article.

Q4: The figures are difficult to understand and in many of the experiments, the necessary controls are lacking.

Answer:

Positive and negative controls were used in the study – following the generally accepted principles of biological in vitro research. The results for both controls are presented in the graphs, and their meaning is explained in the “Experimental Design” section.

Q5: The methods described are not well characterized. How are the cells damaged for example?

Answer:

We have added a detailed procedure for mechanical neurite damage in “Experimental Design” section.

Q6: When testing new biomaterials for peripheral nerve regeneration, it is often tested on primary rat or mice DRG neurons. In this paper cell lines have instead been used. One cell line PC12, that originate from adrenal medulla and bone marrow and SH-SY5Y  from biopsy taken from a four-year-old female with neuroblastoma. These cells do not have a PNS origin and one is of human and one is from a rat origin. Both are cell lines (cancer back ground) and of course regenerate readily. The cell lines are also treated differently and grown on different cell culture surfaces.

Answer:

Of course, the ideal solution would be to use primary neurons. However, the European Directive on the protection of animals used for scientific purposes requires that such studies be approved by the ethics committee on animal experimentation. It is good practice in the biological testing of new compounds to use continuous cell lines as the first step before submitting an application for approval for tests using animal-derived material, including primary lines. Of course, we are aware that the cancer lines used are not the optimal solution. However, in preliminary testing, these cell lines are often used to seek testing on primary and in vivo assay after preliminary studies.

The cell lines were treated differently and grown on different surfaces for their different needs. SH-SH5Y cells are adherent and do not need to have a modified culture surface. By contrast, the PC12 line is growing in suspension. It is well known that in order to create a neurobiological model, culture surface modification is necessary to induce cell adhesion. Only then is it possible to induce the differentiation of the PC12 cell line into neuron-like cells.

Q7: When differentiating cell lines it should be obvious for the reader than they actually turn into neuron like cells, but in this case no such figures or picture of the cells are present.

Answer:

We added microphotographs showing differentiated PC12 and SH-SY5Y cells (Figure 6).

Q8: The entire disposition of the paper needs to be changed. Parts of the results are not results, but more an insurance why this paper is so important! 

Answer:

The article is a combination of chemistry and biology. In such situations, very often, the discussion and results are combined into one chapter. We did the same in this work. Hence, these parts are not “insurance why this paper is so important”, but a discussion of the results in relation to other papers (there are not many articles on this topic yet).

We removed the short paragraph from the beginning of “2.2. Biological Evaluation” that did not fit this section but was a kind of unnecessary introduction.

Q9: Then comes cell culture: Everyone that has grown these two cell lines know that this is not difficult. A bit of antibiotics in the culture is ok, but fungicides? The addition that the slides coated with fibronectin need UV light after coating to presumably kill bacteria? This group really need to work on their sterile technique! Also fungicides often impede cell growth.

Answer:

We respectfully disagree with the Reviewer on this matter. The purchased cell culture plastics are sterile packed by the manufacturer. Their sterility is maintained after opening under the laminar flow cabinet. However, after taking culture plates to the refrigerator (and securing with parafilm) to modify the surface with collagen at 4 °C, it is standard laboratory practice to irradiate such culture vessels with UV – we absolutely cannot agree that this is evidence of our negligence, but it does confirm that we have taken appropriate precautions. Regarding the allegation of the use of fungicides, also, in this case, it is common practice in laboratories to add both antibiotics and amphotericin B in mammalian cell cultures. From the Reviewer's statements, we can assume that he is lucky to work in a monitored clean room – unfortunately, the laboratory reality usually looks different, and the staff must take other additional precautions.

Q10: It is also well known that the two cell lines cannot be passaged for ever. In fact with each passage the proliferation decrease. The paper does not state what the passage was when they started their experiment and when they stopped. In the case of SH-SY5Y cells we have seen a significant decrease in proliferation after 20 passages fx.

Answer:

Of course, we added information about passages in the article in section “3.2.1. Cell Lines”. Passage-dependent morphology variation is also observed in PC12 cells – the more passages, the more their phenotype resembles PC12Adh cells (adherent cell type). In this study, we are not interested in proliferation – that is why we differentiated the cells. At the same time, we remembered and guarded ourselves that the passages were similar throughout the experiment. The differentiation process was carried out on a similar generation of cells.

Q11: The authors are not able to convince me that their biomaterial is really good. How would it work in a nerve repair setting? Would something that is har as enamel really be good? How would it be degraded in the body?

Answer:

Relevant information on this topic has been added to the “Introduction” section of the manuscript.

We hope they will convince Reviewer.

As was reportet by LeGeros [Racquel Z. LeGeros, Biod.egradation and Bioresorption of Calcium. Phosphate Ceramics; Review paper, Clinical Muterids 14 (1993) 65-88], biodegradation or bioresorption of calcium phosphates, including hydroxyapatite, implies cell-mediated degradation and occurs in an acid environment (with acidic enzymes). The release of calcium and phosphate ions from constantly dissolving materials promotes the formation of apatite enriched with these various ions (similar to bone apatite), which increases cell proliferation and activity. This phenomenon is closely related to the bioactivity of the material and its ability to adhere directly to the bone and form a strong material–bone interface. The degradation property of Li+-doped hydroxyapatite was investigated by Wang et al. [Y. Wang, X. Yang, Z. Gu, H. Qin, L. Li, J. Liu, X. Yu; In vitro study on the degradation of lithium-doped hydroxyapatite for bone tissue engineering scaffold, Mater. Sci. Eng C Mater. Biol. Appl., 1 (2016), pp. 185-192] in the context of the possible use of studied biomaterial for bone regeneration. Doping with Li+ ions had a better effect on the growth of osteoblast cells. Our previous studies [Krzysztof Marycz, Paulina Sobierajska, Agnieszka Smieszek, Monika Maredziak, Katarzyna Wiglusz, Rafal J. Wiglusz, Li+ activated nanohydroxyapatite doped with Eu3+ ions enhances proliferative activity and viability of human stem progenitor cells of adipose tissue and olfactory ensheathing cells. Further perspective of nHAP:Li+, Eu3+ application in theranostics, Materials Science and Engineering: C, 78 (2017) 151-162, doi :10.1016/j.msec.2017.04.041] confirmed the literature results about the pro-proliferative and anti-apoptotic effects of lithium on progenitor cells. We showed that lithium ions were effectively released from the apatite matrix, which had a beneficial effect on stem cells, and thus we found a novel Li+ ion delivery method based on the nHAp application.

The cellular mechanism by which the Li+ influences the regeneration of nervous tissue became the subject of many reports. For example, Su et al. [Su, H.; Chu, T.H.; Wu, W. Lithium enhances proliferation and neuronal differentiation of neural progenitor cells in vitro and after transplantation into the adult rat spinal cord. Exp. Neurol. 2007, 206, 296–307, doi:10.1016/j.expneurol.2007.05.018] showed that lithium stimulated the survival, proliferation and differentiation of neural progenitor cells used to treat the damaged spinal cord. Vazey and Connor [Vazey EM & Connor B; Differential fate and functional outcome of lithium chloride primed adult neural progenitor cell transplants in a rat model of Huntington disease. B Stem Cell Res Ther 1 (2010) 41] demonstrated that lithium chloride (LiCl) may improve cell transplantation efficiency in rats and accelerate sensorimotor function recovery. Makoukji et al.  [Makoukji, J.; Belle, M.; Meffre, D.; Stassart, R.; Grenier, J.; Shackleford, G.; Fledrich, R.; Fonte, C.; Branchu, J.; Goulard, M.; et al. Lithium enhances remyelination of peripheral nerves. Proc. Natl. Acad. Sci. U. S. A. 2012, 109, 3973–3978, doi:10.1073/pnas.1121367109] showed that LiCl applied in mice after facial nerve crush injury caused significant stimulation of myelin genes expression, restoration of myelin structure, and thereby recovery of whisker movements. Additionally, Zhu et al. [Zhu Z, Kremer P, Tadmori I, Ren Y, Sun D, He X, Young W. Lithium suppresses astrogliogenesis by neural stem and progenitor cells by inhibiting STAT3 pathway independently of glycogen synthase kinase 3 beta. PLOS One. 6 (2011) e23341] demonstrated that the lithium ions can increase the efficacy of the neural progenitor cells by increasing neurogenesis and reducing astrogliogenesis. This work focuses on the effect of nHAp on the regeneration of mechanical damage in peripheral nerve injuries.

Reviewer 2 Report

The present manuscript reports the effect of two different types of hydroxyapatites (doped with Li+ and Li+/Eu3+) on three different cell types. The study aims to assess the potential regenerative properties of nanohydroxyapatites for neural damage. The study is interesting and provide a well-conducted assessment of the nanomaterial cytocompatibility. However, several comments should be addressed before considering the manuscript for publication:

  1. There is not section 3 in the manuscript.
  2. How did the authors induce the mechanical damage of neuron-like cells? A detailed protocol should be described.
  3. Which type of multi-well cell culture plate did the authors used for the experiments? It is important to know the diameter or area of the well if the number of cells per well is described.
  4. Which type of medium did the authors used for the assays with nanohydroxyapatites, primary medium or differentiated medium?
  5. The authors measured the length of neurites and they detailed that only cells with neurite twice long as length of the body diameter were assessed. Could the authors explain which the approximate percentage of cells analysed per field was? Could authors provide representative images used to quantify the neurite length? It can be included as Supporting Materials.
  6. The results described are difficult to follow. The reviewer suggests to create a short name for both hydroxyapatites in order to simplify the text. In addition, the results are confusing when the authors detailed the concentration ranges of significant differences.
  7. Why did the authors use NHDF cells for cell viability? Why did the authors consider different names for cell viability and mitochondrial activity when two different cell lines were used?
  8. Line 234, Figure 2B should be Figure 6B.
  9. The authors analysed the correlation between MTT assay results and the average neurite length. The results description indicates a “strong correlation” in PC12 and nHAp: 5 mol% Li, 2 mol% Eu. How did the authors calculate it? In addition, values for PC12 are 0.59 and 0.28, whereas for SH-SY5Y are 0.15 and 0.68. The correlation differs according the cell line and hydroxyapatite types. How do the authors explain these results? A discussion should be included.
  10. Line 295, the results are confusing, could the authors clarify the effect of 50 µg/mL concentration on NO levels?
  11. Paragraph starting in line 305 refers to free radicals effect and DNA interaction. Why do the authors hypothesize the interaction with DNA? How could the hydroxyapatite affect the “regeneration” of double-stranded changes? The mechanisms should be detailed in discussion.
  12. Graphs for the average length of neurites are shown in the same style as for MTT. Are the data shown as average length of treated cells/average length of control cells?
  13. A conclusive paragraph or conclusion section should be added at the end of the discussion to summarize the main results.

Author Response

Dear Editor,

We would like to express our sincerest gratitude to the Reviewer for the enormous efforts in criticizing the manuscript. We have considered all raised question here follows the detailed answers to the Reviewer. All changes we have made to the original manuscript are marked in the red color in the text.

Reviewer 2

Comments and Suggestions for Authors:

The present manuscript reports the effect of two different types of hydroxyapatites (doped with Li+ and Li+/Eu3+) on three different cell types. The study aims to assess the potential regenerative properties of nanohydroxyapatites for neural damage. The study is interesting and provide a well-conducted assessment of the nanomaterial cytocompatibility. However, several comments should be addressed before considering the manuscript for publication:

Q1: There is not section 3 in the manuscript.

Answer:

Thank you for this remark. The results section and the discussion have been combined. Hence paragraph 3 has been omitted by mistake. The numbering has been corrected throughout the manuscript.

Q2: How did the authors induce the mechanical damage of neuron-like cells? A detailed protocol should be described.

Answer:

We have added a detailed procedure for mechanical neurite damage in “Experimental Design” section.

Q3: Which type of multi-well cell culture plate did the authors used for the experiments? It is important to know the diameter or area of the well if the number of cells per well is described.

Answer:

96-well plates were used in all biological studies. This information and the density of the seeded cells in all tests have been added in the “Experimental Design” section.

Q4: Which type of medium did the authors used for the assays with nanohydroxyapatites, primary medium or differentiated medium?

Answer:

Primary medium with the tested nanohydroxyapatite was used in biological assays. The aim of the study was to assess the effect of nHaP on neuron-like cells and not on substances that differentiate these cells (NGF, RA) after induction of mechanical damage. Information on this can be found in the section “3.2.3. Tested Compounds”.

Q5: The authors measured the length of neurites and they detailed that only cells with neurite twice long as length of the body diameter were assessed. Could the authors explain which the approximate percentage of cells analysed per field was? Could authors provide representative images used to quantify the neurite length? It can be included as Supporting Materials.

Answer:

Information about the number of measured cells can be found in the section “3.2.7. Length of Neurites”. We cannot say how many cells were measured in the field of view because we measured only those neurites for which we had no doubt that the neurite comes from a given cell and where it ends. In cell cultures, especially PC12 cells, neurite crossings often occur, and it is difficult to distinguish where the neurite ends and from which cell it originates.

We added sample micrographs showing differentiated PC12 and SH-SY5Y cells. We consider it pointless to include more photos used to determine the length of neurites. Due to the nature of the tested parameter, individual photos do not provide any information in this case. A meaningful result can only be obtained after analyzing many photos using, e.g., ImageJ software.

Q6: The results described are difficult to follow. The reviewer suggests to create a short name for both hydroxyapatites in order to simplify the text. In addition, the results are confusing when the authors detailed the concentration ranges of significant differences.

Answer:

Regarding confusing results, we do not fully understand what Reviewer means. Comment Q10 below made a note about the strange results for NO level – we agree that the description was misleading, and we clarified this in response to this note. The influence of the tested compounds was the strongest in the moderate range of the tested concentrations, hence the activity for these concentrations was significant compared to the control.

Q7: Why did the authors use NHDF cells for cell viability? Why did the authors consider different names for cell viability and mitochondrial activity when two different cell lines were used?

Answer:

According to the ISO 10993 standard part V, the cytotoxicity of new compounds should be assessed on fibroblasts. The lines recommended in the standard are of mouse origin. However, human fibroblast lines are increasingly being selected. The fibroblast lines available by us are NHDF (normal human dermal fibroblast). Hence the cytotoxicity was assessed on these fibroblasts.

The MTT test is an indirect test for determining cell viability by assessing mitochondrial activity. Therefore, both terms (“viability” and “mitochondrial activity”) are correct for this assay.

The test on the NHDF cell line had a different purpose than the others. In the case of NHDF cells, they could undergo division, proliferation or cell death, e.g., apoptosis – in this case, we mean the viability. However, in cultures of differentiated cells (PC12 and SH-SY5Y), the process of cell division is stopped. Hence in the tests on these lines, we believe it is safer to talk about the mitochondrial activity (which does not necessarily reflect the viability well in this case).

Q8: Line 234, Figure 2B should be Figure 6B.

Answer:

Thank you. We corrected the number.

Q9: The authors analysed the correlation between MTT assay results and the average neurite length. The results description indicates a “strong correlation” in PC12 and nHAp: 5 mol% Li, 2 mol% Eu. How did the authors calculate it? In addition, values for PC12 are 0.59 and 0.28, whereas for SH-SY5Y are 0.15 and 0.68. The correlation differs according the cell line and hydroxyapatite types. How do the authors explain these results? A discussion should be included.

Answer:

As the obtained results allowed the use of parametric tests, these values were calculated as Pearson correlation coefficients.

The different correlation values may be due to the use of two cell lines of different origins. Even after differentiation, while both are models for neurobiological research, they may have different properties (e.g., dopamine release). Preliminary studies determine whether new compounds have pharmacological potential – and that was the aim of this study. In the next stage, we will delve into the cellular mechanisms and conduct research on primary lines (the most appropriate research model, but requires the approval of the ethics committee on animal experimentation).

We have added in the text of the manuscript a brief comment to different values of these correlations (lines 277-283).

Q10: Line 295, the results are confusing, could the authors clarify the effect of 50 µg/mL concentration on NO levels?

Answer:

All the tests showed that concentrations of nHaP above 20-30 µM negatively affect the cells, i.e., their positive effect was no longer noticeable. At the concentration of 50 µg/mL for nHAp-Li+, the NO level was significantly higher than in control without induced damage, which is not surprising.

Indeed, the text on line 295 was confusing, and the description gave the impression that there was something positive going on at 50 µg/mL concentration, but there is no effect that should be omitted in the description. We have removed this sentence from the manuscript.

Q11: Paragraph starting in line 305 refers to free radicals effect and DNA interaction. Why do the authors hypothesize the interaction with DNA? How could the hydroxyapatite affect the “regeneration” of double-stranded changes? The mechanisms should be detailed in discussion.

Answer:

It is well known that an increase in ROS or NO may increase the number of DNA strand damage. We have not studied the effect of nHaP on DNA damage. This is a plan for the next step. In the papers cited at positions 28 and 30, it can be seen that other authors signal the effect of nHaP on cell DNA. We are not able to say at the moment what mechanism is responsible for this, whether nHaP stimulates intracellular signaling pathways by interacting with membrane receptors or acts only by reducing stress or some intercalation to DNA (least likely). We think it is inappropriate to write more on this topic in the “Discussion" since these are only theoretical hypotheses.

Q12: Graphs for the average length of neurites are shown in the same style as for MTT. Are the data shown as average length of treated cells/average length of control cells?

Answer:

Yes, that's exactly how the data is presented: average length of neurites in treated cells/average length of neurites in control cells. This information was added in the caption of Figure 7.

Q13: A conclusive paragraph or conclusion section should be added at the end of the discussion to summarize the main results.

Answer:

We have added such a paragraph as suggested by Reviewer.

Round 2

Reviewer 1 Report

I still find the methods they have used in cell culture questionable. The only thing that they have done to counter my concerns is adding pictures of the differentiated cells. Here it becomes clear that the SH-SY5Y cells are not appropriately differentiated. 

I am deeply concerned that this paper get another chance after my review. For them to change my mind, they have to completely change their cell culture, using no anti mycotic, the same surfaces for both cell types, using appropriate cells, - real PNS neurons from rats.

Reviewer 2 Report

The authors addressed all the comments, so the reviewer suggest the manuscript for publication.